# Tempo-Spatial Modelling of the Spread of COVID-19 in Urban Spaces

**DOI:** 10.3390/ijerph19159764

**Published:** 2022-08-08

**Authors:** Federico Benjamín Galacho-Jiménez, David Carruana-Herrera, Julián Molina, José Damián Ruiz-Sinoga

**Affiliations:** 1Geographic Analysis Group, Department of Geography, University of Malaga, 29071 Malaga, Spain; 2Department of Geography, University of Malaga, 29071 Malaga, Spain; 3Department of Applied Economics (Mathematics), University of Malaga, 29071 Malaga, Spain; 4Physical Geography and Territory Group, Department of Geography, University of Malaga, 29071 Malaga, Spain

**Keywords:** COVID-19, urban spaces, social areas, socio-spatial structure, statistical analysis

## Abstract

The relationship between the social structure of urban spaces and the evolution of the COVID-19 pandemic is becoming increasingly evident. Analyzing the socio-spatial structure in relation to cases may be one of the keys to explaining the ways in which this contagious disease and its variants spread. The aim of this study is to propose a set of variables selected from the social context and the spatial structure and to evaluate the temporal spread of infections and their different degrees of intensity according to social areas. We define a model to represent the relationship between the socio-spatial structure of the urban space and the spatial distribution of pandemic cases. We draw on the theory of social area analysis and apply multivariate analysis techniques to check the results in the urban space of the city of Malaga (Spain). The proposed model should be considered capable of explaining the functioning of the relationships between societal structure, socio-spatial segregation, and the spread of the pandemic. In this paper, the study of the origins and consequences of COVID-19 from different scientific perspectives is considered a necessary approach to understanding this phenomenon. The personal and social consequences of the pandemic have been exceptional and have changed many aspects of social life in urban spaces, where it has also had a greater impact. We propose a geostatistical analysis model that can explain the functioning of the relationships between societal structure, socio-spatial segregation, and the temporal evolution of the pandemic. Rather than an aprioristic theory, this paper is a study by the authors to interpret the disparity in the spread of the pandemic as shown by the infection data.

## 1. Introduction

In recent years, especially after the emergence of COVID-19, researchers have approached the study of the disease from many perspectives. The different ways in which the pandemic spread is an aspect of great interest. People were affected differently in urban areas, which were the hardest hit during the pandemic, with a complex and diverse distribution. We believe that this was influenced by the varying social circumstances in which different demographic groups live [1]. It should not be forgotten that the urban space can be considered a complex system of interrelated elements, such that a single event can have repercussions overall [2,3].

The purpose of this study was to identify the characteristics of the urban social system that different authors have shown to be the most significant in the context of pandemics and to determine their relationship with officially registered COVID-19 cases [4,5,6,7].

To this end, we focused on the socio-spatial structure of the urban system and located different variables specific to these environments to observe the spatial relationships between them. Accordingly, this model is based on a series of measures and characteristics of the population that are almost universally collected in statistics, at least in EU countries [8,9,10,11,12,13].

Thus, we can propose a method that allows us to compare most European urban spaces or the same city at different times through a process of standardization of these variables. Beyond the specificities of each location, we propose a series of social, family, and territorial variables to define the urban socio-spatial structure, with the aim of explaining the differential behavior of the spread of the COVID-19 pandemic. These variables can be used to propose an interpretation in more universal terms, as conditioning factors common to societies and spaces, even in areas of different cultural or spatial scopes. The possibility of not only obtaining a view of the socio-spatial structure of urban spaces, but also of their potential comparison with other areas is one of the foundations of this paper.

However, it must be assumed that, although all urban spaces share the common denominator of social complexity and heterogeneity, this is directly proportional to the size of the agglomeration and the complex structure of its society [14,15]. This issue is observed in a set of areas of relative internal behavioral homogeneity in relation to the spread of the pandemic, and which have what is called in social psychology, a “sense of community” [16,17].

The basis for the application of the designed model is not only to obtain a view of the socio-spatial structure of urban space, but also to compare it with the data of the COVID-19 pandemic. According to the different authors who have worked on this theory, it is possible to show the essential features by which urban space is structured into socially differentiated areas within the city (social areas). The aim of this is to provide a basis for analyzing the differentiation of society [18,19,20,21].

In summary, having observed the unequal incidence of the COVID-19 pandemic in urban spaces, our aim is to demonstrate that there is a correlation between the variables that shape social and spatial structures and the differential incidence of the pandemic in urban spaces, which is determined both by the living conditions of the population groups and by their characteristics. To demonstrate this hypothesis, we propose a model based on a multidimensional analysis of selected statistical, demographic, and economic data that are summarized in a set of 23 variables.

In essence, these variables have been extracted from the concepts that are considered to assess evolution on the social scale and have a directly observable measure. This proposal of specific indicators ensures that there is a real association within the statistical analysis process. The application of the model will include these correlated variables by means of statistical analysis techniques, as they are conceived as natural associations of multiple tangible and directly measurable characteristics of the populations. Each of them is considered a fundamental aspect of social segregation in the urban space that can be studied separately. Therefore, the model will consider the interactions between the aforementioned variables, which, in turn, are elements of the urban socio-spatial structure. It is the definition of these variables and the establishment of their interrelationships that makes the model useful and will serve as a basis for analyzing the determinants of the distribution of COVID-19 infections in any urban space. We would like to point out that the model will implicitly consider the social inequalities of the different social areas of the urban space, which could be equalized based on the material and social capacities that may be applied as a means of stemming the spread of the pandemic.

Several recent studies have analyzed this fact from similar perspectives. The study of the influence of socio-economic conditions on the living conditions and health of populations is currently being extensively addressed [22,23,24], together with the role of social mobility as a determinant in the spread of the pandemic [25,26,27,28]. Moreover, local and inter-city transmissions of COVID-19 have also been analyzed by examining the role of socio-economic factors and including public health measures by health authorities [29,30].

The urban space of the city of Malaga in Spain was used as an experimental area for testing the proposed model. This is an area of 94.46 km^2^, with a population of 577,405 inhabitants in 2021, according to the official figures of the Population Register of the National Institute of Statistics of the Spanish Government.

## 2. Methods

The starting point for our methodology is going to be the vulnerability index built by Bárcena-Martín et al. (2021) [31], where they measured the level of vulnerability of each census section through a multidimensional vulnerability index. This index was composed of 19 variables arranged in four dimensions: socioeconomic, demographic, care needs of the people who inhabit it, as well as the territorial quality of the space they occupy (Table 1). The selection of variables, described below, among more than 200 originally collected and considered, mostly coincided with those used by other studies and authors for vulnerability analysis in other cities in Spain, as mentioned by the authors in [31]:

This set of 19 variables was composed to obtain a final vulnerability index in another published study [31]. In this paper we have added two: i. The variation between the number of people attended in the Cáritas Spain facilities as an indicator of those areas where the COVID-19 pandemic has caused a higher increase in social care needs (Number 20). ii. The percentage of tourist apartments, as an indicator of the possible effect of touristic mobility on the COVID-19 pandemic (Number 21). 

As done in the original study, a normalization process was applied to all these variables so that all of them were in the range of [0,1], meaning that, in all the cases, 0 was the best situation and 1 was the worst. This is, all the variables were normalized so that in any variable, despite being a maximizing or minimizing one, a higher value means a worse situation in that characteristic. 

Next, the normalized variables were grouped into four categories: demographic, socioeconomic, social care, and territorial, and a linear weighted aggregation was conducted to obtain a final vulnerability index [31]. The authors chose to give the same weight to the dimensions and distribute the weights proportionally within each dimension. That is, they chose to give the same importance to each of the four categories to build the final index. Finally, this index was compared with the COVID-19 data to establish possible relations between the propagation of the pandemic and the underlying social conditions of the population affected. For example, they studied the daily correlation between the vulnerability index and the number of COVID-19 cases in the last 14 days, obtaining this type of curve to be analyzed (Figure 1):

For example, this curve depicts how at the beginning of the pandemic, the higher propagation was found in the less vulnerable areas, as the correlations in this curve at the beginning of the pandemic were negative. 

However, in this paper, we are going to walk the reverse path. Instead of choosing equal, or arbitrary, weights for each of the variables to build the index that will be compared with the COVID-19 evolution, we are going to find the optimal weights so that the correlation with the index built with the COVID-19 data is maximum on each wave (separately, Figure 2), *CO^t^*(*CS_k_*), *t* = 1, …, 4, *k* = 1, …, 434, and for the aggregated data for the whole-time horizon, *CO^f^*(*CS_k_*), *k* = 1, …, 434. As done in the previous work, all these five vectors of COVID-19 data are measured as the number of COVID-19 cases in the last 14 days by 1000 habitants. This is, we are going to find the index using these variables so that its correlation with the number of COVID-19 cases in the last 14 days by 1000 habitants, for each wave separately and aggregated for the whole-time horizon, is maximum (Figure 2).

Formally, we are going to find the vector of weights *W* = (*w_i_*), *I* = 1, …, 21, so that the linearly aggregated index $I_{W}$: $$I_{W}\left(CS_{k}\right)=\sum_{i=1}^{21}w_{i}*VN_{i}\left(CS_{k}\right),\;k=1,\cdots ,\;434$$
is having the maximum linear correlation with the COVID-19 data on the four waves and for the final aggregated data. As a result, we are going to obtain five different sets of weights, *W**_t_*, *t* = 1, …,4 and *W**_f_*, representing the importance of each variable on the spread of COVID-19 through each of the four waves, as well as globally for the whole-time horizon. On the other hand, we allow negative weights to lead to unexpected negative correlations arising. For example, as depicted in Figure 1, on the first wave the less vulnerable populations suffered more COVID-19 spread, so we can expect Income to have a negative correlation with COVID-19 in that wave. This is, we can expect that the more Income the more COVID-19 spread in that wave, so, due to the normalization used (in all cases, 0 is the best situation and 1 is the worst) we can expect a negative correlation between Income and COVID-19, as it finally occurred. So, weights are allowed to lie on the interval [−1,1], instead of the traditional [0,1].

With these definitions, the problems to be solved for each wave and the final aggregated COVID-19 data, are the following:$$\begin{array}{c} \begin{array}{c} max\;Corr\left(I_{W}\left(CS_{k}\right),CO^{t}\left(CS_{k}\right)\right)\\ s.t:\\ W=\left(w_{i}\right),\;w_{i}\in \left[-1,1\right]\;\forall \;i\;in\;\left[1,21\right]\; \end{array}\}\;for\;t=1,\;\ldots ,4\\ \begin{array}{c} max\;Corr\left(I_{W}\left(CS_{k}\right),CO^{f}\left(CS_{k}\right)\right)\\ s.t:\\ W=\left(w_{i}\right),\;w_{i}\in \left[-1,1\right]\;\forall \;i\;in\;\left[1,21\right]\; \end{array} \end{array}$$

These five optimization problems are non-linear, have no explicit formulation, and the objective function is costly to evaluate, so these problems lie in the field of what is considered black box expensive problems. Consequently, to solve them, we chose a specific algorithm based on Scatter Search and Rough Sets [32]. This algorithm was implemented in C++ and was to solve all of them to obtain five sets of vectors, *W**_t_*, *t* = 1, …, 4 and *W**_f_*, representing the importance of each of the 21 variables that explain the COVID-19 spread in the city of Málaga on each of the waves separately and globally as a whole. These sets of optima weights, *W**_t_*, *t* = 1, …, 4 and *W**_f_*, will be analyzed in the following section. 

## 3. Results and Discussion

A total of 34,893 infections were registered in the urban space of the city of Malaga in Spain between 5 March 2020 and 11 November 2021, the distribution of which is shown in the graph below (Figure 3).

Once the temporal evolution of the pandemic is shown, the extent to which the variables analyzed have strengthened or weakened the process of pandemic expansion has to be assessed. It should be kept in mind that the differences between social segmentation and urban structure are implicit in these variables. For this aim, Table 2 shows the importance of each variable on the spread of the COVID-19 pandemic. The first thing to note is the accuracy of the model that we measure with the final correlation coefficients obtained, which are included in the last row of Table 2. All these correlations range over 0.6 and two of them are over 0.7, so the model is shown to be representative. On the other hand, relating to the weights, note that if the weight is above zero, that variable has a positive relationship with the COVID-19 spread. That is, the better an area within that variable is, the less COVID-19 is going to be suffered. Conversely, if a weight is below zero, that variable has a negative relationship with the COVID-19 spread. That is, the better an area is in that variable, the more COVID-19 is going to be suffered. So, for example, in the first wave, the areas where the Caritas variable was increasing the most were suffering more COVID-19 spread (positive weight). However, in the first wave, those areas with a higher home income were also suffering from a higher spread of COVID-19 (a negative weight). Moreover, this negative relationship between household income and the COVID-19 spread is only holding for the first wave. On the following ones, the relation becomes positive (a positive optimal weight), and thus the higher the house income the less the COVID-19 spread. This is not the only variable with a changing relationship with the COVID-19 spread. Similar situations can be found with Dependency Rate, Life Expectancy, Job Seekers, People Served by Social Services, or Home Size. So, this shows how the behavior of the COVID-19 pandemic was complex and changing through the different waves, as it is going to be shown more in depth in the following, using the daily correlation of the most relevant of these variables with the daily number of COVID-19 cases in the last 14 days by 1000 habitants. 

In the structure of the social system, the incidence of these variables takes a concrete form in the configuration of life models, which generate characteristic family situations, thus confirming the existence of alternative family models in urban spaces related to aspects such as the age structure of the population. In short, a set of relationships are shown that highlight differences in the structure of families linked to the life cycle of the populations rather than to the independent ways of life of these age groups. According to their conventional interpretation, these indicators are associated with intergenerational transfers and their systematic increase implies an increase in health-related vulnerability as the population size of the elderly increases. Figure 4a–d show how the daily correlation of the variables of dependency ratio, people over 75 living alone, and the ageing index with COVID-19 cases respond to a similar dynamic, as shown above. Their importance lies in the fact that elderly people living alone tend to have less favorable economic conditions, especially as they get older, which is combined with other contextual aspects such as health and household structure [33]. In people with health problems or sensory impairments, the emergence of new symptoms or the worsening of existing ones may go unnoticed. Many elderly people in these circumstances find it difficult to comply with prescribed treatment regimens. Given their physical limitations and the fact that eating is a social activity, some elderly people living alone do not prepare complete and balanced meals for themselves, which makes malnutrition a common issue in this population [34] and results in increased vulnerability to infectious diseases, as is the case here. In addition, they are considered to be a dependent population. However, in the variable of life expectancy at birth, it can be observed in the last wave how the average number of years lived by the population of the area analyzed, born in the same year, influences age-specific mortality patterns that have been affected by the pandemic during the last wave, which will result in a drop in life expectancy once the disease has been overcome. In fact, in addition to causing a decrease in life expectancy in general, the high mortality from COVID-19 causes a variation depending on the prevailing standard of living in different areas (Figure 4).

Another set of variables shows the change in the order and intensity of relationships in urban spaces. These include those related to the way in which urban dynamics shape the differentiation of economic characteristics throughout a particular space. Four variables have been analyzed here (Figure 5a–d).

We thought that one of the aspects that determine the behavior of the pandemic may lie in social inequality or in the way urban dynamics shape the spatial differentiation of economic characteristics. We considered this aspect from the perspective that the consequences of socio-economic inequalities are transmitted to population health. However, when selecting the variables, we sought to ensure that there was as little mismatch as possible and that the correlation between them was significant enough to be able to expect results that respond to common variance and avoid the problem of redundancy that data on social characteristics inevitably introduce. Thus, it can be observed that one of the most important aspects of the daily correlations in Figure 5 is the variables of job seekers and labor intensity. In the urban areas of Andalusia, the vulnerability of the population has been increasing because of unemployment [35]. As is well known, registered unemployment is a social and economic problem, because it implies high social and financial costs for the state. The importance of this indicator is to be found in the relationship between the degree of labor intensity and the level of well-being attributed to the fact of having a stable job.

This has led us to measure quality of life in relation to disposable income. With this information, it is possible to know the standard of living and living conditions, social cohesion, social protection, and the poverty level of households. Differences in income levels may mean that the incidence of the pandemic decreases when citizens have the possibility of living in less dense areas and further away from congested centers. Social costs have emerged as a result of the different ways of living in the city; households in lower income groups are forced to live in more densely populated spaces, where housing is smaller and cheaper, especially in relation to the number of people to be housed.

The daily correlation observed in the illiterate or uneducated population variable is also noteworthy. As is known, in addition to limiting the full development of people and their participation in society, illiteracy and, mainly, functional illiteracy (people who have basic literacy and numeracy skills but are not able to use these skills efficiently in everyday life situations) has repercussions throughout their life cycle and affects the family environment, restricts access to the benefits of development and hinders the enjoyment of other rights, with important consequences at the personal, intergenerational, social, and economic levels [36].

A series of variables related to the social care perspective, which take a concrete form in the intensity of access to social services, have been included in this model. Fundamental aspects of this are the number of people who resort to social care centers, the detection of social integration needs, and the quantification of the resources applied to subsistence needs. The inclusion of these data in this paper aims to provide information on the capacity of social service centers to alleviate or solve situations of need in two ways: maintaining social care during the state of alert and adapting their social services to the new crisis caused by the pandemic. This analysis has been enriched by assessing the daily correlation with the number of cases, which allows us to confirm that the evolution of the pandemic during the different waves has been directly associated with the social care capacity of public administrations (Figure 6a–d). However, other social agents intervened in this social care framework, such as non-governmental organizations that offered help to the most disadvantaged (Figure 7a,b).

Finally, and after the analysis of the previous variables related to social structure, we will now analyze the urban structure variables (Figure 8a,b). From the analysis of the daily correlation shown in the figure above, we can highlight that the average size of the dwelling was a determining factor. If this is associated with the number of members of the family unit, it can lead to a situation of overcrowding and precariousness that affects the possibility of propagation and the impossibility of complying with lockdown measures [37]. In addition, there are very few studies that propose appropriate mitigation strategies in the design of urban spaces that facilitate the fight against pandemics [38]. The second variable of tourist housing is relevant in the context of pandemic spread. In this case, touristic urban spaces show very favorable conditions for mobility in their residential areas, thus increasing the chances of spread during a pandemic, as can be seen in Figure 8.

The individual analysis of each of the variables carried out herein has the advantage of using the information as much as possible, thus offering many nuances on different aspects of the social areas, which even makes it possible to foresee the areas considered most vulnerable in certain cases.

The results obtained from an analysis of the evolution of the pandemic over time, that is, in the different waves, are very interesting. It is possible to observe that the category that groups variables of demographic type maintains a little fluctuating correlation along the different waves, as can be seen in Figure 4. This leads us to think that it is that at the time of the appearance of the pandemic, the elderly were the most vulnerable. Once the health services were able to react to the care of this group of people, the correlation between the related variables and the cases of COVID-19 is shown without great fluctuations. However, it is important not to forget that the highest mortality was concentrated in the elderly.

Once the spread of the pandemic occurs in a more generalized way, less concentrated in the elderly, the consideration of the category called “social status” becomes important. The correlation between the variables that make up the category and the COVID-19 cases can be seen in Figure 5. We are going to analyze the behavior of the four variables that we have considered most determinant: job seekers, literacy without formal education, labor intensity, and family income level. These variables behave in a very logical, and perhaps expected, way. In the first wave, workers with more stable jobs and necessary mobility accounted for the highest number of infections. Subsequently, it will be the workers in the most precarious situation, who in turn, tend to live in more vulnerable homes, that will suffer the effects of the pandemic, among other effects caused by the urban structure characteristic of the urban space of Malaga. The urban space of the city of Malaga follows a model that we find in many other cities. Basic employment is scattered on the periphery, outside the center, along the main roads and located in industrial areas. However, the services sector is concentrated in the central area. The road network is markedly radial and residential densities vary considerably, being high in and around the center and lower towards the periphery. We have consulted several works in the scientific literature that support this perspective [39,40,41,42].

Differences in the level of income can lead to the incidence of the pandemic decreasing when citizens have the possibility of living in less dense spaces and further away from congested areas. It is noted that social costs appeared to be caused by the different characteristics of the city; the households of the social groups with lower incomes are forced to concentrate in more densified spaces, where housing is smaller and cheaper, especially in relation to the number of people they must house.

It is known that the year 2020 has been highly influenced by the health crisis caused by COVID-19, which, in the field of management of the dependency, has required exceptional measures to contain the contagion. Therefore, the use of parameters that provide information in this regard, we think, is appropriate since they are based on the assessment of the people served. The analysis of the correlation between the variables considered and the COVID-19 cases has been shown in Figure 6 and Figure 7. The comparison of the distribution of cases of contagion with the variables considered in the characterization of the social areas that has been carried out in this work shows that we are in a complex and tertiary society where the particularities of the variables analyzed are appreciable. These point to the existence of clearly differentiated social structures, and there is an obvious relationship between them and the volume of infections, starting from an ageing trend in societies of the type we are dealing with. Thus, the increase in the acute aging of the population is only compensated by the arrival of a new population through immigration. However, although this new population brings youth to the population structure, an effort is also required in the attention that these groups should receive together with the natives of pandemic cases, hence the importance acquired in this work by the variables considered in the category related to social assistance.

## 4. Conclusions

This paper started from the assumption that heterogeneity is one of the differential features of urban spaces described by classical authors to derive or justify the features that define the urban structure and, consequently, the patterns and values of the population groups in social areas [43,44].

The analysis confirms that contrasts in social and family status can be significant in urban spaces, as demonstrated in this particular case, and that differences between social classes can be nuanced and complex and can change according to the level of social care received by social groups. This led to the idea that this heterogeneity results in differential behaviors of the pandemic in different spaces.

A comparison of the distribution of cases of infection with the variables considered in the characterization of the social areas shows that this is a complex society with a predominant tertiary sector, where the specificities of the variables analyzed are apparent. These specificities point to the existence of clearly differentiated social structures, and there is a clear relationship between them and the volume of infections. We can assume there is an ageing trend in societies of this type, and the increase in the acute ageing of the population is only offset by the arrival of new members through immigration. However, although these new members of the population bring youth to the structure of the population, an effort is also required in terms of the social care these groups must receive alongside natives, hence the importance of public actions and investment in the social care framework.

Thus, a key issue is that there is an ambivalence (youth and ageing) in Western societies. This coincidence was observed in variables related to family status. Spatially, this ambivalence acquires greater weight in areas where a lower number of infections have been observed. Talking about young and old spaces is one of the differentiating factors of urban spaces in areas with an individualized social profile. In this case, the areas we have called “city outskirts and areas of recent urban development” show a predominance of the young life cycle. These areas seem to have lower levels of infection compared to the former. The weight of the child or youth population, which has been less affected by the pandemic, means that these areas seem to be less affected. At the same time, these areas are the places of residence or places of residential mobility of people in the middle life cycle: young families with minor children and a higher income level.

The prevalence of older people and their location in areas where they can only have access to small housing units increases their vulnerability to the spread of infections. This group is found in social areas with historical urban development.

By using the selected variables, a multidimensional model was created to classify urban spaces into social areas. With this proposal, we aim to draw attention to some substantial relationships between the social and family structure of spaces and the evolution of the COVID-19 pandemic. On the one hand, the processes that led to the conceptualization of the structure that has been carried out based on the variables and categories analyzed stand out. On the other hand, the processes of the spread of infections and their analysis based on socio-spatial differentiation are highlighted. The mixture of both determines the way in which the pandemic takes place in an urban space, which forces us to classify it into differentiated social spaces. This takes a concrete form in the analysis based on the categories of the model, and the differentiation under this perspective is not only strictly economic—although we have undoubtedly revealed a very close relationship—but also the characterization of the social valuation of space as an indicator of the different possibilities of the social groups in relation to the consequences of the pandemic. The differences observed in this work also show this segregation, which can be explained, in essence, by the ability of some groups (those with higher economic means) to choose the space where they live, while others (those of lower social status) are left with a minimum number of choices, as different studies have shown [45,46,47,48]. Note the difference in the use of terms: the classification of those with higher economic means is determined based on parameters that are more stable, or rather, less vulnerable, while for those of lower social status, more variables besides the economic ones converge and reduce their capacity to choose the space in which they live or make this choice impossible. Therefore, the existence of different opportunities for choice among social classes is a potential source of spatial segregation and indicates a differential behavior of the pandemic.

It should also be noted that a more detailed study of the variables for each wave of infections during the period of the pandemic analyzed has allowed us to delve deeper into the characterization of the different social areas and provide a better explanation of the spread of the virus in the urban space of the city of Malaga in Spain. In other words, we have been able to define the main characteristics of the social areas most affected by the pandemic and how they have evolved over the four waves studied. This information is lost when a summary vulnerability index is applied, as it does not show the casuistry of each variable in relation to COVID-19.

This paper paves the way for a reflection on the configuration of urban spaces as a determinant of the living conditions of populations. This is evident in the different intensities of the spread of the pandemic, which can be observed in social contrasts. Observing the predominance of areas of low social status and an advanced life cycle in these relationships highlights the importance of the intensity of the social differences that can be seen in urban spaces. This suggests that knowing and considering the characteristics of urban society, considering all its areas, can be decisive in the establishment of pandemic control measures.

In order to be able to analyze the characteristics of the spread of diseases in the future, it would be advisable to observe the forms of production of urban spaces together with the social conceptualization of population groups. In this case, we are convinced that the intensity of socio-spatial segregation and the forms of urban structure have determined the intensity of the spread of the COVID-19 pandemic.

The main objective of our work was to analyze the relationship between the COVID spread and the socio-spatial structure, focusing on those social and territorial variables that have influenced the spread of the pandemic the most. This study proposes a set of variables selected from the social context and the spatial structure to be correlated with the temporal spread of infections and their different degrees of intensity according to social areas. The analysis of the correlation between cases and variables will be completed in future work using spatial analysis techniques, as these techniques of spatial analysis have been increasingly utilized to study the territorial impacts of COVID-19. We can currently find a number of tools and strategies to analyze the ongoing pandemic. In this sense, Franch-Pardo et al. (2021) [49] provide a review of 221 scientific articles that used spatial science to study the pandemic published from June 2020 to December 2020. Other authors provide different methods and techniques in this same line [50]. Other aspects of interest that can help to understand the spatial behaviors of the pandemic may be the geographical (or spatial) access to the testing sites (or vaccination locations) that may be related to the spread of the pandemic. In this sense, a series of works have been published [51,52].

So, the future research lines opened devoted to relating social aspects and the structure of cities with the COVID spread have been many, and in this work, we have tried to address some of them. The variables that we have called “territorial” refer to the possible impact that these types of factors can have on the propagation of a pandemic. We have focused the analysis on them, but other very interesting possibilities are opened that will be treated in future works. In this sense, the study of the influence of urban structure is shown in different works as a matter of spatial relevance. We have treated the size of the houses as a determining characteristic and its main use, trying to differentiate between the tourist use as an essential factor for the potential spread of the COVID-19 disease, given the mobility of people (tourists of various origins) that this causes. However, we are aware that there are many other aspects that can be treated and are highlighted in the scientific literature [53,54,55,56,57].

The idea that has been proposed for reflection in this work is to what extent the differences in social segmentation and the current urban structure have supported or weakened the process of expansion of the pandemic. In order to analyze the characteristics of its propagation, it is not possible to isolate the forms of production of urban space from the social conceptualization of space. We are convinced that the intensity of socio-spatial segregation and the forms of the urban structure have determined the intensity of the spread of the COVID-19 pandemic. This type of work, along with many others that have been published in the last two years, can contribute to providing knowledge for decision-making in two aspects: on the one hand, to knowing the impact that the situation of the COVID-19 pandemic has had on hospitals, primary care centers, and social service centers to alleviate or resolve situations of need, in a double direction, maintaining social care in the state of alarm, and adapting their social benefits to new social crises caused by similar situations. On the other hand, in the future, the contributions made by multiple scientific studies that show that the design of cities today is not the most appropriate to address this type of situation should be considered. 

At this point, one of the most important aspects of our work will be to propose the relevant mechanisms for the characterization of urban space according to the mosaic of social worlds in which it is articulated. Its analysis, through the possible delimitations of the spatial differentiation of social groups, is a field of research in which scientific contributions are needed. For example, differences in the level of income can lead to a different incidence of the pandemic if citizens have the chance of living in less dense spaces and further away from congested areas. It is noted that social costs have appeared with origin in the different forms of the city: the households of the social groups with lower incomes are forced to concentrate in more densified spaces, where housing is smaller and cheaper, especially in relation to the number of people they have to house. It is decisive to the extent that the average size of the dwelling in relation to the number of members of the family unit will lead to a situation of overcrowding and precariousness that affects the possibility of spread and the impossibility of complying with the confinement measures [37]. Although, there are very few studies proposing adequate mitigation strategies in the design of urban spaces to facilitate the fight against pandemics [38].

## Figures and Tables

**Figure 1 ijerph-19-09764-f001:**
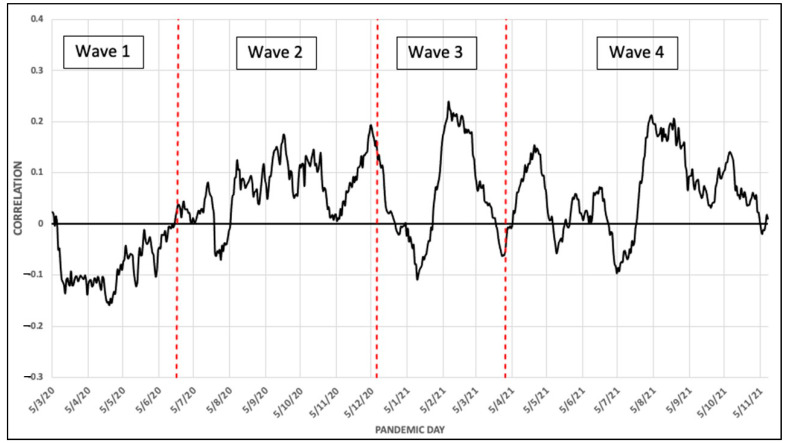
Daily linear correlation between the level of vulnerability and the incidence rate of the census sections of Málaga for the whole-time horizon.

**Figure 2 ijerph-19-09764-f002:**
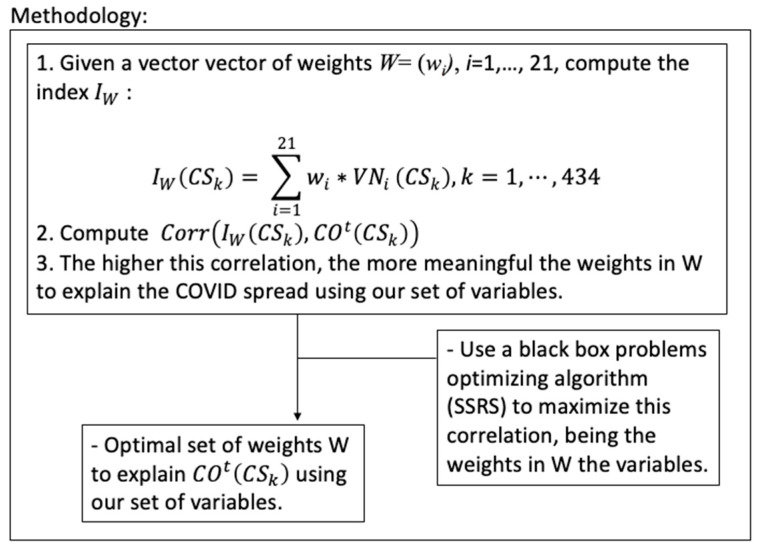
Flowchart of the applied methodological process.

**Figure 3 ijerph-19-09764-f003:**
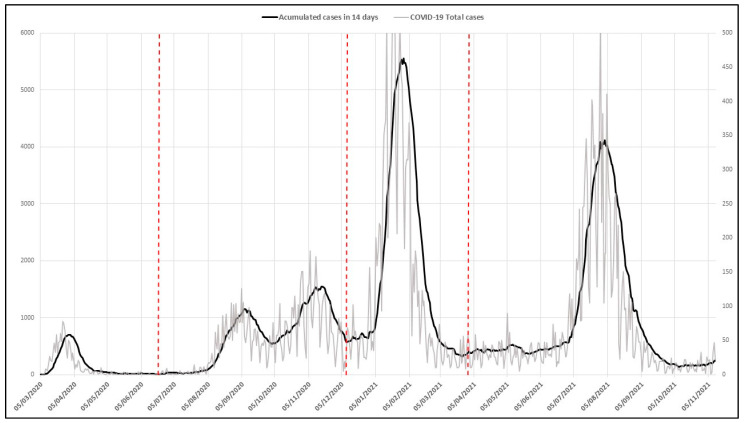
Evolution of the number of COVID-19 infections in the city of Malaga (Spain). Period from 5 March 2020 to 11 November 2021. The vertical red lines show the four waves.

**Figure 4 ijerph-19-09764-f004:**
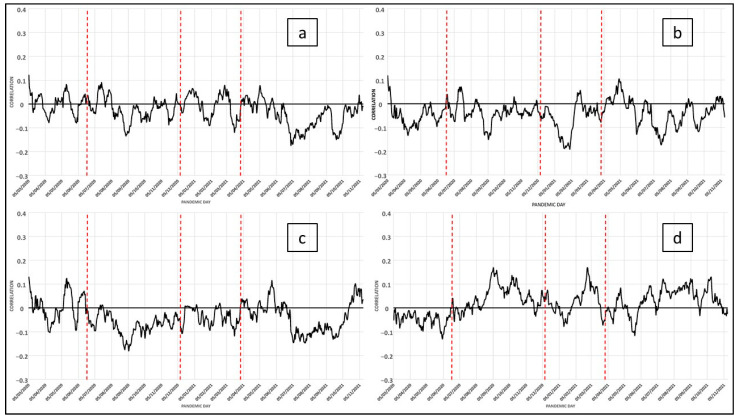
COVID-19: Daily correlation with the number of cases in the last 14 days (for each day). Variables related to family status (demographic category): (**a**), (1) Dependency Ratio; (**b**), (2) People over 75 years old living alone; (**c**), (3) Ageing Index; (**d**), (4) Life Expectancy.

**Figure 5 ijerph-19-09764-f005:**
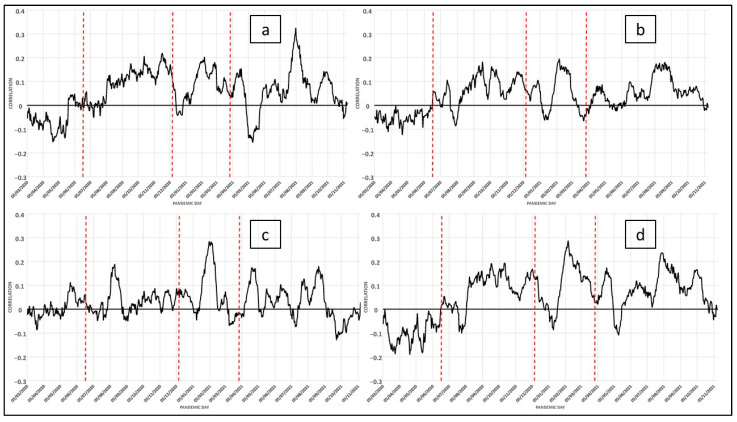
COVID-19: Daily correlation with the number of cases in the last 14 days (for each day). Variables related to social status (socioeconomic category): (**a**), (7) Job Seekers; (**b**), (6) Literacy without formal education; (**c**), (8) Labour Intensity; (**d**), (5) Family Income Level.

**Figure 6 ijerph-19-09764-f006:**
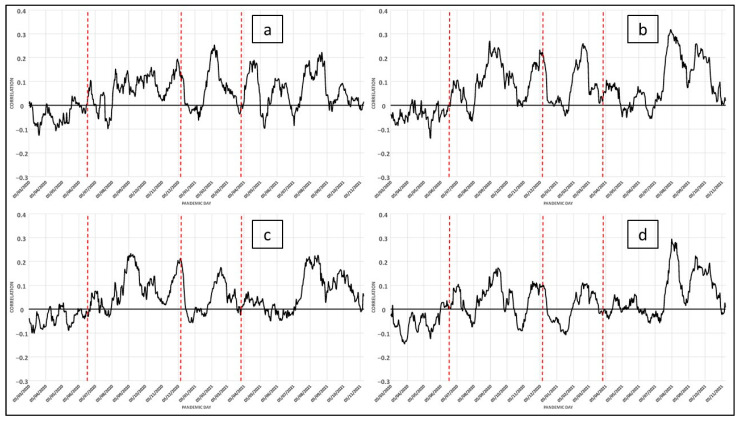
COVID-19: Daily correlation with the number of cases in the last 14 days (for each day). Variables related to social care: (**a**), (10) People Served by Social Services; (**b**), (9) No Material Deprivation; (**c**), (11) Social Integration Needs Detected; (**d**), (12) Resources Applied to Subsistence Needs.

**Figure 7 ijerph-19-09764-f007:**
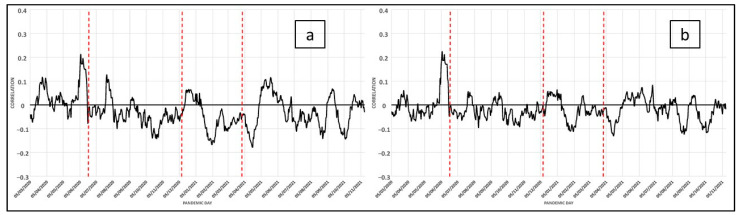
COVID-19: Daily correlation with the number of cases in the last 14 days (for each day). Variables related to NGOs’ social care: (**a**), (20) NGOs’ Social Care. Pre-pandemic; (**b**), (20) NGOs’ Social Care. Post-pandemic.

**Figure 8 ijerph-19-09764-f008:**
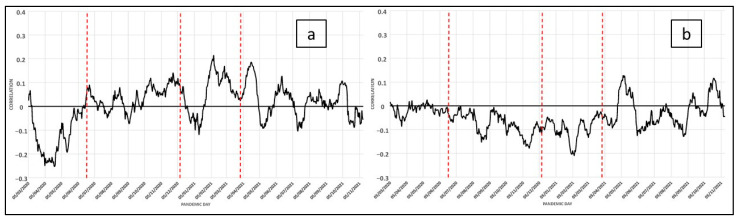
COVID-19: Daily correlation with the number of cases in the last 14 days (for each day). Variables related to urban structure: (**a**), (19) Urban Structure: Average Home Size; (**b**), (21) Tourist Housing.

**Table 1 ijerph-19-09764-t001:** Variables used for the construction of the vulnerability index.

	Variable	Definition	Unit	Source	Date
Demographic	(1) Dependency	Persons under 16 and over 64 years of age with respect to the total active population	%	Municipal Register, OMAU	2019
	(2) Ageing	Persons over 64 years of age compared to those under 16 years of age	%	Municipal Register, OMAU	2019
	(3) 75+ alone	Persons over 75 who live alone out of the total population	%	Municipal Register, OMAU	2019
	(4) Life expectancy	Average age reached by the population	Years	Municipal Register, OMAU	2019
Socioeconomic	(5) Household income	Average annual net income of households (set of income received minus taxes and social security contributions)	Thousand euros	INE, OMAU	2017
	(6) Illiterate or uneducated population	Percentage of the population over 16 years of age that is illiterate or has no education	%	INE (Census)	2011
	(7) Job seekers	Percentage of the population between 16 and 65 years of age registered with the public employment services to search for a job or for other purposes	%	SEPE, Municipal Register	Dec. 2019
	(8) Work intensity	Percentage of household members willing to work who work	%	Survey	2019
	(9) No severe material deprivation	Constructed index that indicates the percentage of the population that lives in households that can afford at least six items out of a ratio of nine	%	Survey	2019
Welfare	(10) People served	Percentage of people served by community social service centres over the total population	%	SIUSS	2019
	(11) Social integration needs detected	Percentage of assessments made by community social services professionals on social integration needs presented by users	%	SIUSS	2019
	(12) Resources applied to subsistence needs	Percentage of resources applied from community social service centres to meet subsistence needs of the population served	%	SIUSS	2019
Territorial	(13) Green zones	Total green areas per inhabitant, square meters per inhabitant	Square meters per inhab.	OMAU	2019
	(14) Altitude	Own elaboration from E:1.10.000 and from the centroid of each neighbourhood meters	meters	Topographic map	2018
	(15) Orientation	Own elaboration from the MDT of Malaga with ArcGis	Degrees longitude	Topographic map	2018
	(16) Torrentiality	Incidence of large downpours	Rate	AEMET	2012
	(17) Differences on the maximum temperature	Own elaboration, through a field study based on a citizen science experiment	Degrees °C	Own elaboration	2013
	(18) Accessibility	Index built from a group of proximity variables	% population	OMAU	2019
	(19) Average size of the dwelling	Average size of the dwellings calculated from the size of the dwellings of the alphanumeric data of the cadastre	Square meters	OMAU. Cadastre	2020

**Table 2 ijerph-19-09764-t002:** Weights of variables at four time points (waves) of the pandemic.

Var.	Wave1	Wave2	Wave3	Wave4	Final
(1) Dep Rate	−0.138	0	0.7188	0	−0.5606
(2) 75+Alone	0	0.2784	−0.7016	−0.5236	0
(3) Ageing	−0.663	−0.265	0	−0.855	−0.5877
(4) LifExp	−0.3978	0	0	0.468	0.0128
(5) HousInc	−0.1324	0.7749	0.2798	0.6951	0.3864
(6) Illiterate	0	0	0.8642	0.4294	0.3862
(7) JobSeek	−0.9179	0.9915	0.1588	−0.0621	−0.1967
(8) WorkInt	−0.1154	0	0.9325	−0.2401	0.0296
(9) NoSevDep	0	0	0.8625	0.6583	0.899
(10) PeopServ	−0.5864	0.5281	0	0	0.5591
(11) SocIntNeeds	−0.6669	0.3737	0.0119	0	0.8369
(12) NecSub	0	−0.3773	−0.2207	0	0.5002
(13) GreenZon	0	−0.3854	0	0	0.6778
(14) Altitude	−0.7803	−0.4487	0	−0.0074	0.4026
(15) Aspect	−0.2769	−0.6299	0	0	−0.1042
(16) Torrenc	0.8391	0.9279	0	−0.1132	0.7266
(17) DifTMax	−0.3154	−0.252	−0.5051	0	−0.898
(18) Accessib	0.5754	−0.6656	0.6002	0.0328	−0.2536
(19) HomeSize	−0.8907	0.0299	0	0	−0.5839
(20) NGOs’SocCare	0.9594	−0.3652	0	0	−0.6207
(21) TourApart	0.0103	0.7899	0	0	0.7361
Correlation	0.6368	0.7428	0.6478	0.6666	0.7355

Legend: Final correlations obtained and optimal weights for each variable on each pandemic wave (Wave*_i_*) and for the whole-time horizon aggregated (Final). To know about the acronym of each variable see Table 1.

## Data Availability

The datasets analyzed during the current study are public. The COVID-19 databases are in the process of being authorized for publication.

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
