# Peer review of "Tempo-Spatial Modelling of the Spread of COVID-19 in Urban Spaces"

_ijerph, 2022, doi:10.3390/ijerph19159764_

Round 1
Reviewer 1 Report
Major Comments
1. The first paragraph in Results and Discussion section needs to be included in the part of data as the study area, which denotes “The urban space of the city ~~~~”.
2. Just looking at the correlation between the cases and other variables may not that helpful. You could find other methods in the following references. Although you may not interested in performing new analysis, at least you could mention the methodological limitations.
Franch‐Pardo, I., Desjardins, M. R., Barea‐Navarro, I., & Cerdà, A. (2021). A review of GIS methodologies to analyze the dynamics of COVID‐19 in the second half of 2020. Transactions in GIS, 25(5), 2191-2239.
Jiao, J., Chen, Y., & Azimian, A. (2021). Exploring temporal varying demographic and economic disparities in COVID-19 infections in four US areas: based on OLS, GWR, and random forest models. Computational urban science, 1(1), 1-16.
3. In many papers regarding the COVID-19, it is shown that geographical (or spatial) access to the testing sites (or vaccination locations) may be relative to the COVID-19 spreads. You could also mention these in the paper.
Kang, J. Y., Michels, A., Lyu, F., Wang, S., Agbodo, N., Freeman, V. L., & Wang, S. (2020). Rapidly measuring spatial accessibility of COVID-19 healthcare resources: a case study of Illinois, USA. International journal of health geographics, 19(1), 1-17.
Kim, K., Ghorbanzadeh, M., Horner, M. W., & Ozguven, E. E. (2021). Identifying areas of potential critical healthcare shortages: A case study of spatial accessibility to ICU beds during the COVID-19 pandemic in Florida. Transport Policy, 110, 478-486.
Minor comments
1. It would be better to merge Figure 1 and Figure 2 together.
Author Response
Thank you very much for your fruitful comments, that drove us to a better version.
Please see the attachment.

Reviewer 2 Report
The paper is well written and has certain social significance. I have, however, several concerns:
The title of this article shows that it is related to spatial modeling, but it is not well presented in this paper. It is more like a temporal modeling.
In the Results and discussion part, there should be added some explanation of the accuracy of the model, so as to determine whether the model is suitable for this study and judge the credibility of the model by the accuracy.

Author Response

(The authors gave the same response as above.)

Reviewer 3 Report
The topic presented in the paper is current and interesting, the paper is well articulated and written, so minor revision are required before the publication.
The themes included in the Introduction are very interesting, however, reference should also be made to how the spread of the COVID-19 pandemic is related to home spaces, their sizing, the degree of crowding in residences and also to the impact on the rental prices and on the housing market in general. In this sense there are numerous studies applied to the Italian context.
In section 2 Methods, in order to make all the steps of the proposed method more understandable, it is advisable to add either a graphic diagram or a list summarizing the main steps. It is also important to describe how the data collected have been processed to test their statistical robustness.
In Section 3 Results and discussion it is advisable to add a small paragraph tended to summarize the outcomes for each of the categories of the selected variables (there is also a typo In the acronym covid instead of covid-19).
In Section 4 Conclusions it is advisable to explain what decision-making processes the research carried out could support and what spin-offs it can have in terms of city design.
Author Response

(The authors gave the same response as above.)

Round 2
Reviewer 2 Report
It would be neccessary to add some indexes to testify the accuracy of the model in the Results and discussion part, so as to determine whether the proposed model is suitable for this study and judge the credibility of the model by the accuracy.
Author Response
- It would be neccessary to add some indexes to testify the accuracy of the model in the Results and discussion part, so as to determine whether the proposed model is suitable for this study and judge the credibility of the model by the accuracy.
In the previous version, we included some indexes to testify the accuracy of the model in Table 2. More in detail, we included de correlation coefficients for each pandemic wave and the whole aggregated data (Final), as specified in the (modified) legend of this table. But we didn’t include any explanatory comment in the “Results and discussion” section, so now we did in, to underline the meaning of these coefficients for the accuracy of the model [page 219]
Please see the attachment.
